# The Accuracy of 3D Surgical Design and Simulation in Prefabricated Fibula Free Flaps for Jaw Reconstruction

**DOI:** 10.3390/jpm12111766

**Published:** 2022-10-26

**Authors:** Sherif Idris, Heather Logan, Paul Tabet, Martin Osswald, Suresh Nayar, Hadi Seikaly

**Affiliations:** 1Division of Otolaryngology—Head and Neck Surgery, Department of Surgery, University of Alberta, Edmonton, AB T6G 2B7, Canada; 2Institute for Reconstructive Sciences in Medicine, Misericordia Community Hospital, Edmonton, AB T5R 4H5, Canada; 3Division of Otolaryngology-Head & Neck Surgery, Department of Surgery, Université de Montréal, Montreal, PQ H1T 2M4, Canada

**Keywords:** personalized medicine, dental osseointegrated implants, prefabricated fibula free flap, mandible and maxilla surgery, surgical design and simulation, virtual planning, additive manufacturing

## Abstract

The ideal jaw reconstruction involves the restoration and maintenance of jaw continuity, jaw relations, joint alignment, and facial contour, and, most importantly, dental occlusal reconstruction. One of the essential requirements of achieving a consistent functional outcome is to place the bony reconstruction in the correct three-dimensional position as it relates to the other jaw segments and dentition. A protocol of occlusion-driven reconstruction of prefabricated fibular free flaps that are customized to the patient with surgical design and simulation (SDS)-planned osseointegrated implant installation was developed by our institution. This innovation introduced significant flexibility and efficiency to jaw reconstructions, but functional and cosmetic outcomes were dependent on the accuracy of the final reconstructions when compared to the SDS plan. The purpose of this study was to examine the accuracy of the SDS-planned fibular flap prefabrication in a cohort of patients undergoing jaw reconstruction. All patients that had undergone primary jaw reconstruction with prefabricated fibular free flaps were reviewed. The primary outcome of this study was the accuracy of the postoperative implant positions as compared to the SDS plan. A total of 23 implants were included in the analysis. All flaps survived, there was no implant loss postoperatively, and all the patients underwent all stages of the reconstruction. SDS planning of fibular flap prefabrication resulted in better than 2 mm accuracy of osteointegrated implant placement in a cohort of patients undergoing jaw reconstruction. This accuracy could potentially result in improved functional and cosmetic outcomes.

## 1. Introduction 

The ideal jaw reconstruction involves the restoration and maintenance of jaw continuity, jaw relations, joint alignment, and facial contour, and, most importantly, dental occlusal reconstruction. One of the essential requirements of achieving a consistent functional outcome is to place the bony reconstruction in the correct three-dimensional (3D) position as it relates to the other jaw segments and dentition. This occlusion-driven reconstruction of the jaws was first described in 2003 by Rohner et al. [1]. 

The use of surgical design and simulation (SDS) in head and neck reconstruction has increased over the past decade. Surgery is planned virtually and, once completed, the SDS-planned reconstruction is translated back to a physical plan that can be implemented in the operating room using various tools and guides. Studies have demonstrated that SDS can reduce surgical time, ischemia time, and inaccuracies compared to analog planning [1,2,3,4,5,6,7,8]. SDS-assisted reconstructions maintain 3D spatial relationships of the reconstruction and are essential when reestablishing dental occlusion with osseointegrated implants [9,10,11]. 

An occlusion-driven reconstruction protocol of prefabricated fibular free flaps with SDS-planned osseointegrated implant installation was developed by the Division of Otolaryngology—Head and Neck Surgery at the University of Alberta and the Institute of Reconstructive Sciences in Medicine in Edmonton, AB, Canada [12]. This innovation introduced significant flexibility and efficiency to reconstructions. We also developed surgical tools which, in combination with a set of instruments and components (FIRST System, Southern Implants, Irene, South Africa), allowed the transfer of the SDS plan as a custom, fully guided resection and reconstruction procedure for the patient in the operating room.

The accuracy of the final reconstructions when compared to the SDS plan is essential for achieving a functional occlusion and cosmetic outcomes [12]. The purpose of this study was to examine the accuracy of the SDS modifications of fibular flap prefabrication in a cohort of patients undergoing jaw reconstruction. 

## 2. Material and Methods

### 2.1. Study Design

The University of Alberta Health Research Ethics Board approved this study on 26 November 2019. (Pro00096288). The cohort of patients were followed prospectively. 

### 2.2. Patients

All patients that had undergone primary jaw reconstruction with prefabricated fibular free flaps were reviewed. Six had completed their treatment protocol and were included in the final analysis.

### 2.3. Virtual Surgical Planning

Each patient had a high-resolution helical Computed Tomography (CT) scan of the facial bones and the fibula. Images were stored in an uncompressed Digital Communications in Medicine (DICOM) format. These files were imported into Mimics Medical 17.0 software (Materialise, Leuven, Belgium) and were then segmented and reconstructed into 3D digital models.

The virtual surgical planning was carried out using Geomagic Freeform software (3D Systems, SC, USA) during a web-based online planning session between the primary reconstructive head and neck surgeon, a maxillofacial prosthodontist, and a surgical simulationalist at the Medical Modeling Research Laboratory at the Institute for Reconstructive Sciences in Medicine, Misericordia Community Hospital, Edmonton, AB, Canada. During this planning session, the resection planes of the jaw were established based on the surgeon’s clinical judgment. Participants in the planning session were able to simultaneously view 3D representations of the patients’ anatomy, plan optimal implant positions based on the patients’ native dentition, perform virtual resections of the jaws, and plan the fibular reconstruction position based on the planned implant positions (Figure 1).

Using the virtual surgical plan, a patient-specific fibular implant guide was fabricated for the Stage I surgery, and additional patient-specific surgical guides were fabricated for the Stage II surgery. Specifically, a fibula osteotomy guide, patient-specific reference models of the fibula, presurgical models, planned reconstruction models, and mandible or maxillofacial resection cutting guides, were created for Stage II. All models were manufactured using 3D printers and sterilized for surgical use.

### 2.4. Surgical Procedure

#### 2.4.1. Stage I: Flap Prefabrication

The surgical implant drilling guide was mounted on the fibula in the predetermined position, and the osseointegrated dental implants were instilled. Next, an impression (with dental impression materials) of the final positions of the implants was taken to aid in the design of future dental prostheses. The fibula and implants were then covered with a split-thickness skin graft and a Gore-Tex^®^ patch (Preclude^®^ Dura Substitute, W.L. Gore & Associates, Inc., Flagstaff, AZ, USA) as described by Rohner et al. [1] (Figure 2).

#### 2.4.2. Stage II: Jaw Reconstruction

Six months after Stage I, the jaw resection and fibula reconstruction were performed. The fibula was re-exposed and removal of the Gore-Tex^®^ membrane revealed the newly attached epithelial tissue around the implants and along the lateral sides of the fibula (Figure 3A). The flap was elevated and placed into a fibular holder (Southern Implants, Irene, South Africa) to maintain vascularization and safe manipulation of the flap during surgery (Figure 3B). The surgical cutting guide was repositioned on the implants, and the fibula was further osteotomized as indicated by the patient-specific fibular cutting guide produced by SDS based on the preoperative virtual surgical plan (Figure 3C). Proper configuration of the bone segments according to the planned digital design was achieved by mounting the interim dental prosthesis on the osseointegrated implants (Figure 3D) 

The flap and transfer template construct were relocated to the jaw, and the patient was placed in occlusion, with the construct in the accurate spatial relationship. The bone segment(s) of the flap were plated to the jaw using mini plates. Finally, soft tissue adjustment and microvascular anastomoses were performed. 

#### 2.4.3. Stage III: Prosthodontic Treatment

The definitive dental prosthesis was fabricated and delivered after healing was complete (Figure 4). 

### 2.5. Primary Outcome

The primary outcome of this study was the accuracy of the postoperative implant positions as compared to the SDS plan. Each patient had a high-resolution helical CT scan of the facial bones using either a Somatom Sensation (Siemens, Germany) or a GE VCT (GE Healthcare, Waukesha, WI) 64-slice CT select scanner via a 0.625 mm collimation with a 25.0 cm field of view and 0 degree gantry tilt. All patients had postoperative CT scans of their facial bones 6–12 months after the Stage II procedure. 

### 2.6. Data Analysis

The preoperative digital plan, referred to as “planned”, and the scans of the postoperative results, referred to as “actual”, were used for our analysis. Digitally placed spheres (1 mm diameter) were manually positioned in the geometric center of the planned and actual implant positions along the occlusal surface of the fibula. The spheres defined the reference point for measuring implant position. The *X*, *Y*, and *Z* coordinates of each planned and actual implant position were obtained (Figure 5 and Figure 6). The difference in position of the dental implants between the preoperative planned and postoperative actual CT scans was calculated in millimeters. 

## 3. Results

### Patients

Six consecutive patients were included in the analysis. The patient demographics are shown in Table 1. A total of 24 dental implants were inserted, 23 of which were included in the statistical analysis. One implant was inserted at the time of the first stage of surgery but was not planned using SDS and was excluded from our analysis. All flaps survived, there was no implant loss postoperatively, and all the patients that underwent all stages of the reconstruction maintained functional and stable occlusion. 

When the postoperative scans were superimposed on the preoperative SDS plans, the mean center-point distances between the actual and planned implant positions were 1.5 mm (SD ± 1.2 mm) in the *X* axis, 2.0 mm (SD ± 1.0 mm) in the *Y* axis, and 1.8 mm (SD ± 1.1 mm) in the *Z* axis (Figure 7). 

## 4. Discussion 

This study examined the accuracy of the SDS workflow and AM modifications of fibular flap prefabrication in a cohort of patients undergoing jaw reconstructions. The mean deviations in implant positions from the virtual surgical plans were 1.5, 2.0, and 1.8 mm in the *X*, *Y*, and *Z* axes, respectively. Our findings suggest that the use of SDS in planning osseointegrated dental implants in the prefabricated fibula achieves spatially accurate results in patients undergoing jaw reconstruction. In addition, the minor deviations seen in our cohort were not clinically significant, as they could easily be accommodated for in the final prosthesis. Specifically, all patients in the study developed and maintained good functional occlusion. 

Our findings are supported by evidence in the literature. Schepers et al. [13] evaluated the accuracy of placement of fibular grafts and dental implants compared to a virtual plan during a one-stage procedure involving the immediate installation of dental implants. They found a mean deviation of 3.3 mm between the virtually planned implants and the postoperative implants [13]. 

The use of vascularized bone free flaps has improved the functional and aesthetic outcomes of osseous reconstructions [14,15]. The fibula flap most similarly resembles the jaw, both dimensionally and biomechanically [16,17]. This similarity is why we chose to assess the fibula free flap for our study and is why the fibula free flap is used in our routine practice in caring for patients requiring reconstruction of the jaw for similar defects. The fibula provides excellent bone stock, good soft tissue components, and has a long pedicle that has good caliber vessels for anastomosis [18,19]. The fibula is composed of strong bicortical bone, offering increased primary stability to an implant that is superior to other donor sites [14,15,17,20,21,22,23,24]. Furthermore, the bone can be further augmented by impacting the marrow of the donor fibula with demineralized bone or morselized fibular bone, resulting in improved implant longevity [25,26].

This study reports the clinical accuracy of SDS for osseointegrated implant positioning in prefabricated fibular free flap procedures. We showed that the use of SDS and personalized patient plans improve accuracy of osteointegrated implants and potentially may improve the patients’ functional and cosmetic outcomes after jaw reconstruction. The study had some limitations because it was a single-center study and, therefore, susceptible to biases of such a design. Furthermore, the sample size was small so the results must be interpreted with caution. 

## 5. Conclusions

SDS planning of fibular flap prefabrication resulted in better than 2 mm accuracy of osteointegrated implant placement in a cohort of patients undergoing jaw reconstruction. This accuracy could potentially result in improved functional and cosmetic outcomes 

## Figures and Tables

**Figure 1 jpm-12-01766-f001:**
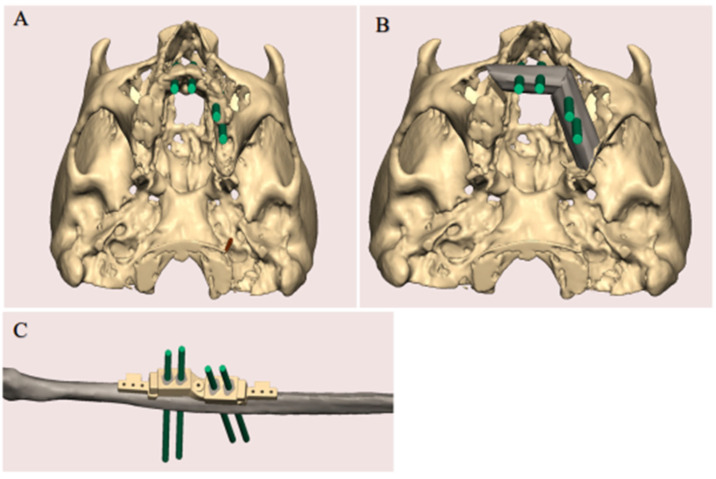
SDS planning. (**A**) Optimal implant positions based on the patient’s native dentition were confirmed. (**B**) The resection planes of the jaw were established. (**C**) The position of the fibula reconstruction based on the planned implant positions.

**Figure 2 jpm-12-01766-f002:**
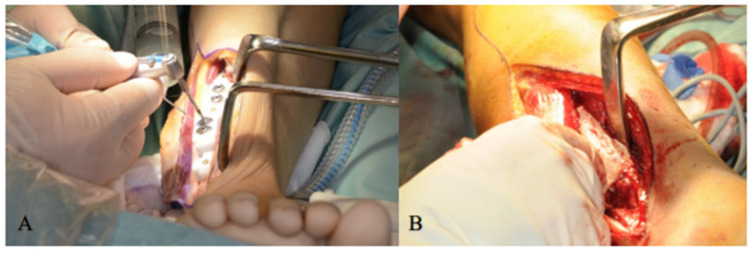
(**A**) Three-dimension printed drilling guide is anchored to the fibula and osseointegrated implants are placed in the planned location. (**B**) Split-thickness skin graft placed over the lateral aspect of the fibula containing the implants.

**Figure 3 jpm-12-01766-f003:**
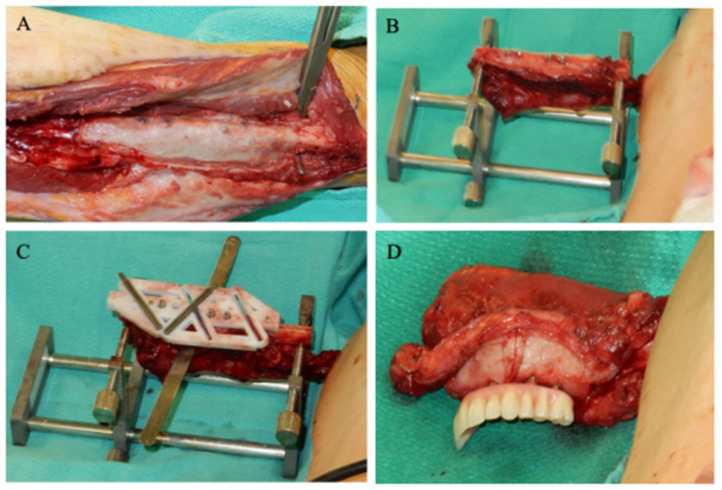
(**A**) Fibula was re-exposed, and removal of the Gore-Tex^®^ membrane revealed the newly attached epithelial tissue around the implants and along the lateral sides of the fibula. (**B**) The flap was placed into a fibular holder (Southern Implants, Irene, South Africa) so that the flap could be safely manipulated during surgery while remaining vascularized. (**C**) The surgical cutting guide was repositioned on the implants, and the fibula was further osteotomized. (**D**) Proper configuration of the bone segments was established, with the interim dental prosthesis on the osseointegrated implants. The construct was relocated to the jaw and the flap was plated to the jaw using mini plates.

**Figure 4 jpm-12-01766-f004:**
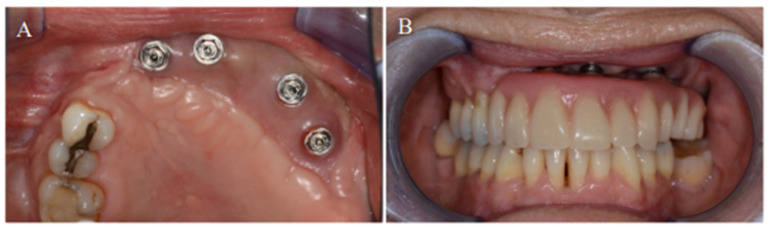
(**A**) Complete osseointegration of the dental implants into the neomandible. (**B**) Placement of the final acrylic dental prosthesis).

**Figure 5 jpm-12-01766-f005:**
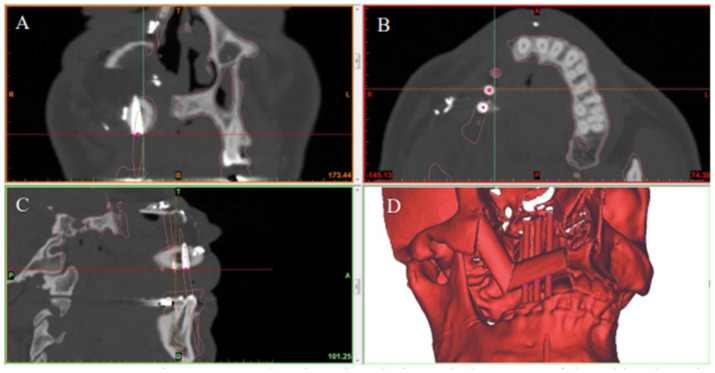
Preoperative CT scan showing virtual planned placement of dental implants in the reconstructed jaw in the (**A**) coronal, (**B**) axial, and (**C**) sagittal planes. (**D**) 3D model showing the final jaw reconstruction with implants in the ideal position according to virtual surgical plan.

**Figure 6 jpm-12-01766-f006:**
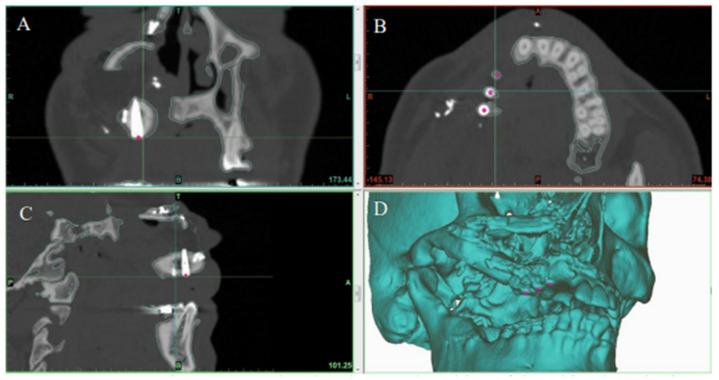
Postoperative CT scan showing the actual position of dental implants in the reconstructed jaw in the (**A**) coronal, (**B**) axial, and (**C**) sagittal planes. (**D**) 3D model showing final jaw reconstruction with implants in their actual position.

**Figure 7 jpm-12-01766-f007:**
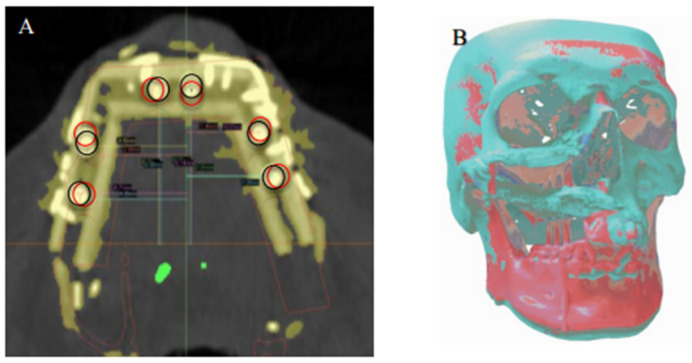
(**A**) Superimposition of the planned (black circles) dental implant locations according to virtual reconstruction planning and actual (red circles) implant locations on the postoperative CT. (**B**) Registration of the preoperative 3D model and the postoperative 3D reconstruction, showing overall deviation from the virtual plan.

**Table 1 jpm-12-01766-t001:** Demographics.

Patient	Age (Years)	Sex	Diagnosis	Jaw Defect
1	46	F	Ameloblastoma	Maxilla
2	27	M	Hemangioma	Maxilla
3	61	M	SCC	Maxilla
4	23	M	Ameloblastoma	Mandible
5	49	M	Keratocyst	Maxilla
6	56	F	ORN	Maxilla

F: female; M: male; ORN: osteoradionecrosis; SCC: squamous cell carcinoma.

## Data Availability

The data presented in this study are available on request from the corresponding author. The data are not publicly available due to privacy and ethical restrictions.

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
