# Peer review of "The Accuracy of 3D Surgical Design and Simulation in Prefabricated Fibula Free Flaps for Jaw Reconstruction"

_jpm, 2022, doi:10.3390/jpm12111766_

Round 1

Reviewer 1 Report

This is an excellent paper, I enjoyed reading about your technique and experience. I did not see any figures or illustrations, which I believe would be exceptionally beneficial and add to the content you presented in the manuscript. 

Author Response

Thank you for your review. I have uploaded the images with the revised manuscript

Reviewer 2 Report

My compliments for your jobs.

A mandibular defect classification is necessary to analyze the problem (Pavlov, Urken, Cordeiro, Rodriguez, Brown, Valentini-Della Monaca etc).

A different analyze about SCC patient needed in term of timing and complication

Author Response

Thank you for you comments.

I agree that a mandibular defect classification is important but most of the defects were maxillary in nature. A full mandibular defect classification discussion is beyond the scope of this paper

The patient with SCC was reconstructed in the same timeliness the other patients and did not have any complications. 

Reviewer 3 Report

This papers is aimed to examine the accuracy of the surgical design and simulation modifications of fibular flap prefabrication in a cohort of patients undergoing jaw reconstructions.

Over the last years, many protocols have been developed and tested in this matter. Authors fail to provide a strong rationale to develop and test another protocol.

The title is misleading, as the accuracy is a one-point measure of the implants. What about the implant's angles? What about the contour of the fibula?

In general terms is a succint, well-written paper. But only provide one-point linear measure. As they have the "plan" and "actual" images, I strongly advised to enrich the results with another measures.

Author Response

Thank you for your comments and insightful suggestions.

We developed this pilot study to evaluate the accuracy of the surface of the implant in the x, y, and z dimensions. We are involved in another international study to develop a comprehensive measurement rubric.

We did not specifically evaluate the of the implant angulation because the interm prothesis in the second stage locked the fibular segments together and placed them in occlusion guaranteeing proper angulation. We evaluated the contour of the reconstruction through the scans as detailed in figure 7 but were not able to provide a meaningful objective measure.

Round 2

Reviewer 2 Report

After revision, the manuscript is ready for publication

Reviewer 3 Report

Authors have included some minor modification across the manuscript. However, the main issue was no addressed, this is a small case series reporting only 3 linear measurements. I hope they will include a comprehensive evaluation in the international study they are taking part.